# Hierarchical Patch VAE-GAN:
# Generating Diverse Videos from a Single Sample

**Shir Gur**\*, **Sagie Benaim**\*, **Lior Wolf**
The School of Computer Science, Tel Aviv University

## Abstract

We consider the task of generating diverse and novel videos from a single video sample. Recently, new hierarchical patch-GAN based approaches were proposed for generating diverse images, given only a single sample at training time. Moving to videos, these approaches fail to generate diverse samples, and often collapse into generating samples similar to the training video. We introduce a novel patch-based variational autoencoder (VAE) which allows for a much greater diversity in generation. Using this tool, a new hierarchical video generation scheme is constructed: at coarse scales, our patch-VAE is employed, ensuring samples are of high diversity. Subsequently, at finer scales, a patch-GAN renders the fine details, resulting in high quality videos. Our experiments show that the proposed method produces diverse samples in both the image domain, and the more challenging video domain. Our code and supplementary material (SM) with additional samples are available at https://shirgur.github.io/hp-vae-gan.

## 1  Introduction

Video is often considered an extension of images, and in many cases, methods that are applied to images, such as CNNs, are also applied to video sequences, perhaps with 3D convolutions instead of 2D ones. However, when considering synthesis, video generation introduces challenges that do not exist for images. First, one needs to generate, not just one, but many images. Second, video generation requires accurate continuity in time, especially since the human visual system easily spots violations of continuity. Third, the composition of the scene needs to make sense both with regards to object placement, and with regards to their motion and interactions.

A fourth challenge, that has become apparent during the development process, is that the problem of mode collapse is much more severe in videos than in images. Recent GAN-based approaches for synthesis from a single-sample have shown that variability is readily created by hierarchical patch modeling. However, in video, each patch is constrained along three different axes, and the temporal axis does not necessarily comply with the fractal structure of images [1, 2, 3]. Therefore, a direct generalization of such models to video results in a generation that collapses to the original video.

To overcome this, we first offer a novel patch-VAE formulation that explicitly models the patch distribution of a video. This approach enables the faithful reconstruction of each patch in the video, as well as the generation of novel samples, thus avoiding the problem of mode collapse and memorization of the input video sample. We subsequently use our patch-VAE in a novel hierarchical formulation. The newly formulated hierarchical patch-VAE employs a multi-scale decoder but only one encoder. The VAE's $\mathbb{KL}$ constraint is only applied to the activation map that this single encoder outputs. The different scales of the decoder are trained sequentially, such that the encoder and the top part of the decoder are trained at each step with a reconstruction term.

---

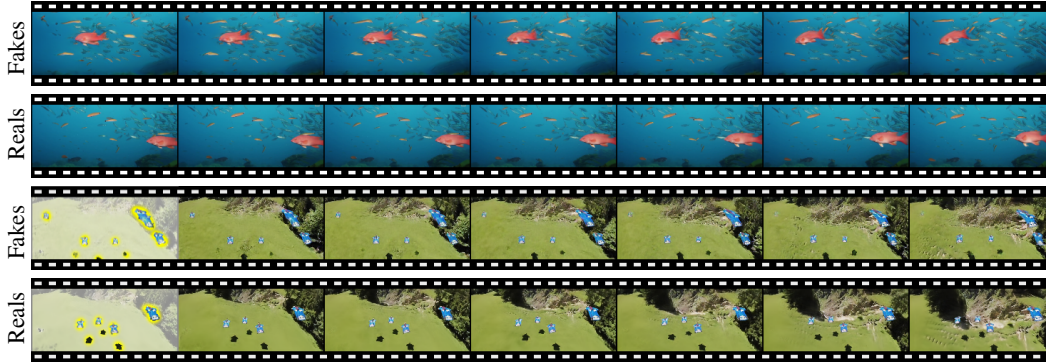

Figure 1: Random video generations. Showing every other frame for both reals and fakes. **Row 3 and 4 -** Highlighted on first frame are the skydivers. Note the difference in composition and amount. Full videos are available in the supplementary material (SM) .

The sequence of VAE decoder modules at multiple scales stops before the top resolution, at which point, multiple scales of conditional GANs are used to upsample the video. The two generator types (VAE and GAN) play different roles, and determining the number of scales for each of the two types has a dramatic effect on the results. The role of the patch-VAE is to generate diverse samples at coarse scales. At these scales, the global structure of the video, including the background and the various objects and their placement, is determined. At finer scales, the role of the patch-GAN is to refine those samples by adding fine textural details, resulting in high-quality samples. The alternative of using patch-GAN for all levels of the hierarchy results in low diversity, and samples resembling the input video. On the other extreme, using patch-VAE for all levels results in low quality samples. This is shown experimentally in section Sec. 4. A sample result of our framework is shown in Fig. 1.

Our experiments show that the novel method outperforms previous work, where we compare to (i) current video generation models, which can only be trained on multiple video samples, and thus benefit from an unfair advantage, (ii) recent image generation methods trained on a single image sample, and (iii) the extension of these methods to video, which, does not replicate their success in image generation. Finally, we consider the effect of different components of our method, such as the number of patch-VAE levels, the receptive field, and updating networks only at specific layers.

To summarize, our work provides the following novelties: (1) a new patch-VAE formulation for a single sample generation, which encourages large diversity and avoids mode collapse, (2) a novel patch VAE-GAN method, which generates diverse and high quality samples, and (3) the first method capable of unconditional video generation from a single video sample.

## 2 Related Work

**Multiple Samples Generative Models** The use of generative models for novel content synthesis has been a topic of significant research [4, 5, 6, 7, 8, 9]. In the context of images, VAEs [10, 11, 12], pixelCNNs [13], pixelRNNs [14] and their variants have proven effective in modeling the underlying probability distribution of the data, while faithfully reconstructing and generating novel samples. GANs [4, 5], while not modeling the explicit probability distribution, have shown impressive ability in generating high quality samples. To further improve the quality of generated samples Karras *et al*. [6] proposed a hierarchical model that progressively increases the resolution, at which a GAN is trained. Other work, such as StyleGAN [7], LAPGAN [15] and StackGAN [16] also employed a mutlti-scale generation process. Several hybrid models were proposed in an attempt to combine the benefits of GANs and VAEs [17, 18, 19, 20]. However, these typically do not operate within a hierarchy framework. A recent work by Gupta *et al*. [21] proposed a patch-based VAE (PatchVAE). However, unlike the one we employ, it cannot be trained on a single sample.

**Single Sample Generative Models** Several GAN-based approaches were proposed for learning from a single image sample. Deep Image Prior [22] and Deep Internal Learning [23], showed that a deep convolutional network can form a useful prior for a single image in the context of denoising, super-resolution, and inpainting. However, they cannot perform unconditional sample

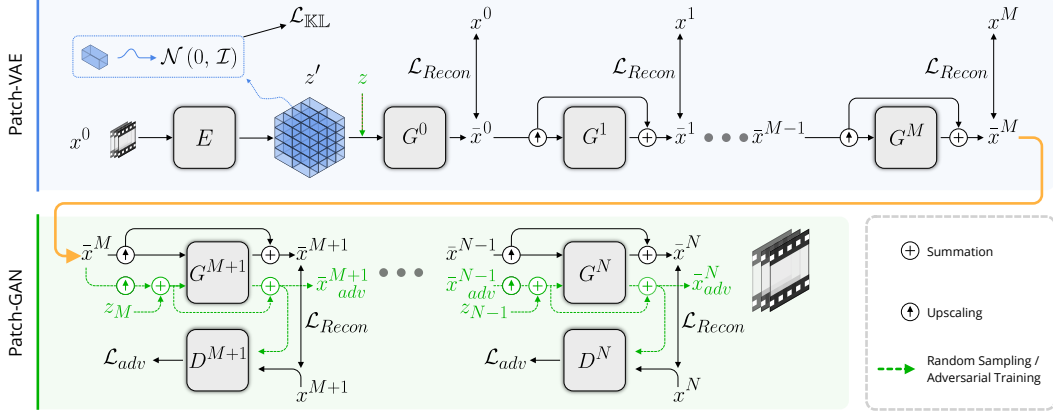

Figure 2: Our Hierarchical Patch VAE-GAN framework. The model takes as input a video sample in low resolution, or a latent vector $z$. First, a hierarchical patch-VAE is trained to create high variability in lower scales, and second, a hierarchical patch-GAN is trained to obtain high quality outputs.

generation. Our work is inspired by that of SinGAN [24], which uses patch-GAN [25, 24, 26, 27] to model the multiscale internal patch distribution of a single image, thus generating novel samples. ConSinGAN [28] extends SinGAN, improving the quality and train time. As we show, both SinGAN and ConSinGAN suffer from mode collapse when applied to video.

**Video Generation** Villegas et al., [29] investigate the minimal inductive bias required for video prediction tasks. Wu et al., [30] introduce a novel approximation of the sliced Wasserstein distance, resulting in superior video generation performance. VGAN [1] employs GANs by separately generating the foreground and background of a video. TGAN [2] decomposes the spatiotemporal generator into a temporal generator and an image generator. MoCoGAN [31] decomposes the video latent space into motion and content latent spaces. TGAN-v2 [3] proposed a differentiable sub-sampling layer which reduces the dimensionality of intermediate feature maps. Acharya *et al*. [32] proposed a progressive-like architecture for video generation. Similarly to our method, both the spatial and temporal resolution is increased as training progresses. DVD-GAN [33] leverages a computationally efficient decomposition of the discriminator to produce high quality samples. Unlike our method, all of these approaches employ a significant number of samples, and have a large memory footprint.

## 3 Method

We first outline our novel patch-based variational autoencoder (VAE) architecture and training details. We then describe our hierarchical patch-based generation scheme, which we coin hierarchical patch VAE-GAN (hp-VAE-GAN). In the context of this work, we describe the method for 3D generation. The required adjustments for image generation are straightforward.

### 3.1 Patch-VAE

The standard VAE framework [10] receives as input i.i.d samples $x$ and a prior distribution $p(z)$ over a latent space $z$. It then learns the conditional distribution $p(x|z)$ and a variational approximation to the intractable posterior distribution, denoted $q(z|x)$. $p(x|z)$ and $q(z|x)$ are often realized by:

$$z \sim E(x) = q(z|x), x \sim G(z) = p(x|z) \tag{1}$$

where $E$ and $G$ are neural networks. $G$ is considered as a decoder (or generator) and $E$ as an encoder. $E$ and $G$ are learned to minimize the following variational lower bound (ELBO):

$$\begin{aligned} \mathcal{L}_{\text{VAE}}(x) &= -\mathbb{E}_{z \sim q(z|x)} \left[ \log p(x|z) \right] + \mathbb{KL} \left[ q(z|x) \parallel p(z) \right] \\ &= -\mathbb{E}_{z \sim E(x)} \left[ \log G(z) \right] + \mathbb{KL} \left[ E(x) \parallel p(z) \right] \end{aligned} \tag{2}$$

In our problem setting, we do not receive multiple samples $x$, and instead, receive a single sample $x$. The distribution we model is of the patches of $x$. $E$ is a fully convolutional encoder, with an effective

receive field $r$, and we consider all $r$-sized (possibly overlapping) patches of $x$. Each such patch, $\omega$, is drawn from the distribution of appropriately-sized patches that $x$ entails, which we denote as $\mathbb{P}_x^r$.

**Encoding**  Given the input video $x$, $E(x)$ is an activation map of size $T \times H \times W \times C$, where $T$, $H$ and $W$ denote the time, height and width dimensions, and $C$ is the number of channels. In a complementary view, $E$ can be seen as creating a distribution $q(z|x)$ of the latent encoding vector $z \in \mathbb{R}^C$ of $r$-sized patches, a distribution from which we have $K := T \times H \times W$ samples. We denote the individual encoding vectors as $z_i$, and each is associated with a patch $\omega_i$. We therefore consider the $\mathbb{KL}$ loss $\sum_{i=1}^K \mathbb{KL}\left[z_i \parallel p(z_i)\right]$, where $z_i$ is the $i$'th entry of $z$ and $p(z_i)$ is the prior distribution for latent space of patch $\omega_i$. As is common with the VAE formulation, the prior $p(z_i)$ is assumed to be a multi-variate normal distribution $\mathcal{N}(0, \mathcal{I})$. The $\mathbb{KL}$ loss is minimized by using the reparameterization trick. For this purpose, the mean $\mu_i$ and standard deviation $\sigma_i$ are estimated for each patch $\omega_i$. This is done by using two, dimension preserving, convolution layers, $M$ and $S$, and having $\mu_i = M(z_i)$ and $\sigma_i = S(z_i)$. With this reparameterization, the above $\mathbb{KL}$ loss becomes:

$$\mathcal{L}_{\mathbb{KL}}(x) = \sum_{i=1}^K \mathbb{KL}\left[\mathcal{N}(\mu_i, \sigma_i) \parallel \mathcal{N}(0, \mathcal{I})\right] \tag{3}$$

**Decoding**  Assuming that $q(z|x)$ of Eq. 2 is a Gaussian, $-\mathbb{E}_{z \sim q(z|x)}\left[\log p(x|z)\right]$ becomes the reconstruction error between x and $G(z)$. In the patch based setting, we would like to reconstruct $\omega_i$ given $z_i$. And so one could sample $z_i' = \epsilon\sigma_i + \mu_i$ where $\epsilon \sim \mathcal{N}(0, \mathcal{I})$, and apply a reconstruction loss $\sum_{i=1}^K \|\omega_i - G(z_i')\|_2$. We suggest to employ a decoder $G$ that is fully-convolutional as well, and apply it to the vector field $z'$ obtained by reassembling the $z_i'$'s into the shape of the activation map $E(x)$, putting each $z_i'$ into the position corresponding to the center of $\omega_i$. This way, the generated patches $G(z_i')$, are not independent given $E(x)$, and are generated such that the overlapping "generative fields"[2] are consistent. This is illustrated in Fig. 3 for the 2D case. The reconstruction loss is then:

$$\mathcal{L}_{\text{Recon}}(x) = \|x - G(z')\|_2 \tag{4}$$

As is done in $\beta$-VAE [11], a parameter $\beta$ weights the $\mathbb{KL}$ loss term, resulting in the loss:

$$\mathcal{L}_{vae}(x) = \mathcal{L}_{\text{Recon}}(x) + \beta\mathcal{L}_{\mathbb{KL}}(x) \tag{5}$$

When devising the architecture for $E$ and $G$, one needs to set $r$ not too large with respect to the input dimensions of $x$, so that $K$ corresponds to a large enough sample space. On the other hand, setting $r$ too small may correspond to only finer textural details of $x$. Further implementation details are provided in the SM. Note also, that our formulation naturally extends to multiple samples $x$, or augmentations of $x$, by summing the loss of Eq. 5 over all such samples. This is in contrast SinGAN, which requires reconstructing each sample, at the the coarsest level, from the same fixed noise $z^*$, resulting in bad reconstructions. This property is of great importance when training on long videos, in which case, we can cut the video to multiple fixed-sized samples.

## 3.2  Hierarchical Patch VAE-GAN

We now describe our novel hierarchical approach for generating diverse video samples. We employ $N+1$ scales, where 0 is the coarsest scale, and $N$ is the finest scale. Fig. 2 illustrates our framework.

The role of the first $M$ scales is to generate structural diversity, *i.e.*, modify the scene composition, including the number of objects and their positions. VAEs can produce highly diverse samples and are not prone to mode collapse [10, 18, 19], making them good candidates for this task. Our experiments in Sec. 4 show that using patch-GAN instead of patch-VAE results in considerably less diversity.

From scales $M+1$ onward, the receptive fields become smaller, and the top-level generators introduce fine textural details. At these scales, we wish to encourage quality over diversity, which patch-GAN does well [4, 5, 6, 7, 8]. We show experimentally in Sec. 4 that using a patch-VAE for all scales, results in diverse, yet blurry and low quality videos.

When using a patch-GAN, we follow a similar procedure to SinGAN [24], in using a fully convolutional generator and discriminator of a fixed effective receptive field $r$, while varying the (both temporal and spatial) resolution of $x$ at each scale. The same technique is used for the patch-VAE modules, which allows us to use the same architecture across different generator types and scales.

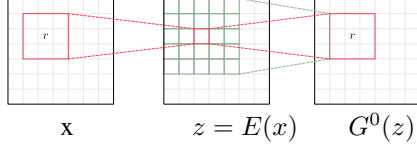
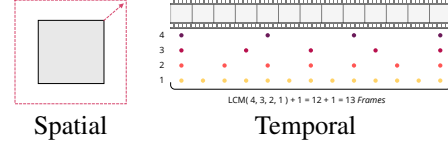

| x | $z = E(x)$ | $G^0(z)$ |
|---|---|---|

| Spatial | Temporal |
|---|---|

Figure 3: 2D illustration of the latent space receptive field (RF). **Red -** RF of a single $z^i$. **Green -** Region around $z^i$ which influence patch $\omega_i$

Figure 4: Sampling and interpolating in the spatio-temporal domains. The figure illustrates temporal sampling rates of 1, 2, 3 and 4 frames.

**Sampling and Interpolating**  For each scale $n = 0, .., N$, $x$ is down-sampled both spatially and temporally to create $x^n$. Spatially, we perform bilinear interpolation keeping the aspect ratio of $x$. Temporally, we reduce the video frame rate (FPS) by uniformly sub-sampling video frames, resulting in frames that are less subjected to motion blur or pixel interpolation. We assume a set of sampling rates $\Omega$. Scale 0 (resp. $N$) uses the highest (resp. lowest) value of $\Omega$ and the value at $n$ is chosen by linearly indexing between the possible values of $\Omega$. To ensure the start and end frames remain the same, as the FPS changes, we choose slices of size $LCM(\Omega) + 1$ ($LCM$ is the least common multiple). We use a 13 frame sequence from the original video, taken at 24 FPS, and sub-sample accordingly, as illustrated in Fig. 4. When temporally and spatially up-sampling a video, we use trilinear interpolation, ensuring that the first and last frames do not change over time. We set $\Omega$ to $\{1, 2, 3, 4\}$ in our experiments.

**The first $M$ scales**  We start training at scale 0, where we use a patch-VAE with encoder $E$ and decoder $G^0$, and train our patch-VAE using $x^0$ as described in Sec. 3.1. We denote the reconstructed sample at this stage as $\bar{x}_0$. $E$ and $G^0$ (as are all the $G^i$'s) are chosen to preserve the input dimensions (using padding), see SM for details. This scale minimizes the loss $\mathcal{L}_{vae}(x^0)$ as defined in Eq. 5 and is trained for a fixed number of epochs.

Each of the subsequent scales $n = 1, .., M$ is trained based on the networks of the previous scales. At scale $n$, the encoder $E$ and the generator $G^0$ are fine-tuned, and $G^n$ is initialized from $G^{n-1}$ and trained. $G^1, .., G^{n-1}$ are not updated (frozen).

The generator at scale $n = 1, .., N$ (this is relevant for the patch-GAN scales as well) is trained to provide a residual signal. Let $\bar{x}^{n-1}$ be the output of the previous scale, and $\uparrow \bar{x}^{n-1}$ be the result of upsampling $\bar{x}^{n-1}$ to the scale of level $n$. We define $\bar{x}^n$ to be:

$$\bar{x}^n = \uparrow \bar{x}^{n-1} + G^n(\uparrow \bar{x}^{n-1}) \qquad (6)$$

Thus, each generator progressively adds detail to the upscaled version obtained from the previous generator. The reconstruction loss at scale $0 < n \leq N$ is given by:

$$\mathcal{L}_{\text{Recon}}(\bar{x}^n, x^n) = \|\bar{x}^n - x^n\|_2 \qquad (7)$$

Although the encoder's input is of low resolution, $E$ is still updated through $x_n$'s reconstruction loss, allowing the latent space to capture $x_n$'s finer details. The loss minimized at scale $n$ is then:

$$\mathcal{L}_{vae}(x^0, \bar{x}^n, x^n) = \mathcal{L}_{\text{Recon}}(\bar{x}^n, x^n) + \mathcal{L}_{\text{Recon}}(\bar{x}^0, x^0) + \beta_{vae}\mathcal{L}_{\mathbb{KL}}(x^0) \qquad (8)$$

The $\mathbb{KL}$ term of this loss pertains to the encoder $E$, while the others affect $E$, $G^0$, and $G^n$. Updating $E$ and $G^0$ allows the entire network to adapt to the highest resolution. $G^{n'}$ for $0 < n' < n$ are not updated, and only exist to serve the final resolution.

**Scales $n > M$**  From scale $M + 1$ onward, our method employs a patch-GAN for each scale, training a generator $G^n$ and discriminator $D^n$. As stated, $G^n$ is trained in a residual manner, similarly to Eq. 6, learning to add details to samples from the previous scale. $D^n$ produces a single channel spatio-temporal activation map of the same dimension as its input, indicating whether each $r$-sized patch of the input is real or fake. In particular, we first sample $z \in \mathcal{N}(0, \mathcal{I})$ of the same shape as $E(x^0)$. We then apply Eq. 6 to the $M$ scales patch-VAE generators $G^0,\ldots,G^M$ and obtain $\bar{x}^M$.

For $n > M$, two different outputs are computed during training. One is $\bar{x}^n$, which is obtained using the recursion of Eq. 6. The other $\bar{x}^n_{rand}$ is obtained using randomness, and is defined as follows:

$$\bar{x}^n_{rand} = \begin{cases} \uparrow \bar{x}^{n-1}_{rand} + G^n(\uparrow \bar{x}^{n-1}_{rand} + z_n) & n > M \\ \uparrow \bar{x}^{n-1}_{rand} + G^n(\uparrow \bar{x}^{n-1}_{rand}) & 0 < n \leq M \\ G^0(z') & n = 0 \end{cases} \qquad (9)$$

where $z_n$ is a random noise of the same shape as $\uparrow \bar{x}_{rand}^{n-1}$ and $z'$ is a random noise sampled from the prior distribution as described in Sec. 3.1. This recursion allows to produce random samples and, in particular, different objects and details. The adversarial loss is given by the WGAN-GP [34] loss:

$$\mathcal{L}_{adv}(z, x^n) = \min_{G^n} \max_{D^n} \mathbb{E}[D^n(x^n)] - \mathbb{E}[D^n(\bar{x}_{rand}^n)] - \lambda \mathbb{E}[(\|\nabla_{\bar{x}_{rand}^n} D^n(\bar{x}_{rand}^n) - 1\|_2)^2] \quad (10)$$

where $\mathbb{E}$ is mean over $D^n$'s output. $\bar{x}_{rand}^n$ is a randomly generated sample at scale $n$ and $\bar{x}_{rand}^n = \epsilon x^n + (1 - \epsilon)\bar{x}_{rand}^n$ for $\epsilon$ sampled uniformly between 0 and 1. In addition to the adversarial loss, we also employ a reconstruction loss as in Eq. 7. The overall loss at step $n > M$ is:

$$\mathcal{L}_{adv}(z, \bar{x}^n, x^n) = \mathcal{L}_{\text{Recon}}(\bar{x}^n, x^n) + \beta_{adv}\mathcal{L}_{adv}(z, x^n) \quad (11)$$

for some $\beta_{adv} > 0$. In the patch-GAN training, i.e., for $n > M$, only $G^n$ and $D^n$ are trained, while $E$ and $G^0, \dots, G^{n-1}$ are frozen. This is unlike the first $M$ layers, in which $E$ and $G^0$ are fine-tuned. Unlike [24, 28], our model naturally extends to multiple video samples by dividing a longer video into multiple parts. We simply sum the above-mentioned loss terms over all such samples. During inference, we sample a noise vector $z \sim \mathcal{N}(0, \mathcal{I})$ and apply Eq. 9 up to the final scale $N$.

## 4 Experiments

We begin by comparing our method to current video generation methods, which employ a large number of samples. Next, we follow the intuitive extension of current single-sample image generation, to video, and evaluate our method with these baselines. We also demonstrate the advantage of our method when applied to images. We consider the effect of training with different numbers of VAE levels $M$ on the diversity and realism of the generated samples. We also consider the effect of the receptive field $r$ on our patch-VAE, and the effect of freezing part of the layers during training. Unless otherwise stated, we set $M = 3$ and $N = 9$ and use the architecture described in the SM.

**Datasets** To compare to video generation methods, we use the UCF-101 dataset [35], which contains over 13K videos of 101 different sport categories. For single sample video experiments, we choose 25 high quality video samples from the YouTube 8M dataset [36]. For a single sample image experiment, 25 images were randomly selected from SinGAN's training samples[3].

**Single Video FID** We adapt the Single Image FID metric introduced in SinGAN [24] to a Single Video FID metric (SVFID) by using the deep features of a C3D [37] network pre-trained for action recognition instead of an Inception network [38]. As shown in SinGAN [24], SIFID is a good measure for how fake samples look, compared to the real sample trained on. See further details in the SM.

### 4.1 Multiple Sample Video Generation Baselines

We consider the baselines of MoCoGAN [31], TGAN [2] and TGAN-v2 [3] (Acharya *et al.* [32] and DVD-GAN [33] did not provide code or generated samples), trained on the UCF-101 dataset [39]. We randomly sample 50 generated videos and for each sample $s$, find their 1st and 2nd nearest neighbors (NN) in the UCF-101 training set, denoted $nn_1$ nad $nn_2$. We then train our model on each of the $nn_1$ samples separately and generated a random sample $s'$ for each $nn_1$ sample. When finding the 2nd nearest neighbor, we verified that this is from a completely different video. To evaluate each method, we apply the SVFID metric between the baseline's sample $s$ and each of $nn_1$ and $nn_2$ and repeat this for our $s'$. Comparing to the first is biased in favor of our method, since we train on this sample; comparing to $nn_2$ is biased against our method since our method did not see this sample during training and $nn_2$ was selected based on $s$. Aside from this, the protocol is fair, since using one sample out of the training set is a valid training strategy. We do not claim to generate variable content from multiple videos, as may be the case for the baselines. However, our method is superior to baselines in generating diverse content from the same internal statistics of the input video.

Tab. 1 compares the SVFID score of our method and the baselines. Due to the experiment design, different NN samples are used for each baseline, and the comparison is only valid for each baseline separately and not across baselines. As can be seen, our method results in a superior SVFID score (both in comparison to $nn_1$ and $nn_2$), indicating that our results are more realistic. Sampled results are provided in the SM. Using the 50 generated samples $s$ or $s'$ we also computed an FID score against UCF-101 test set. As can be seen in Tab. 2, our method has superior FID score then baselines indicated more realistic samples. FID and SVFID implementation details are provided in the SM.

Table 1: Mean SVFID for each method, w.r.t.1st and 2nd NN. Each column represent a set of 50 samples extracted for each method as described in Sec. 4.1, to which our method was compared.

| | MoCoGAN's samples [31] | | TGAN's samples [2] | | TGAN-v2's samples [3] | |
|---|---|---|---|---|---|---|
| NN | 1st | 2nd | 1st | 2nd | 1st | 2nd |
| Baseline | 19.0 | 19.1 | 20.0 | **20.6** | 25.4 | 25.5 |
| Ours | **6.1** | **15.3** | **8.7** | 21.0 | **8.9** | **20.6** |

Table 2: Mean FID over UCF-101 test samples. Each column represents 50 samples extracted for each method, to which our method was compared.

| | MoCoGAN's samples [31] | TGAN's samples [2] | TGAN-v2's samples [3] |
|---|---|---|---|
| Baseline | 42.0 | 28.3 | 29.2 |
| Ours | **22.1** | **16.1** | **17.2** |

Table 3: User study Mean Opinion Scores (1-5) for (C1): Realness and (C2): Diversity.

| | Videos | | Images | |
|---|---|---|---|---|
| | C1 | C2 | C1 | C2 |
| SinGan [24] | 3.6 | 2.9 | 2.3 | 2.5 |
| ConSinGan [28] | 3.4 | 2.4 | 2.5 | 1.8 |
| Ours | **4.1** | **3.8** | **3.5** | **3.6** |

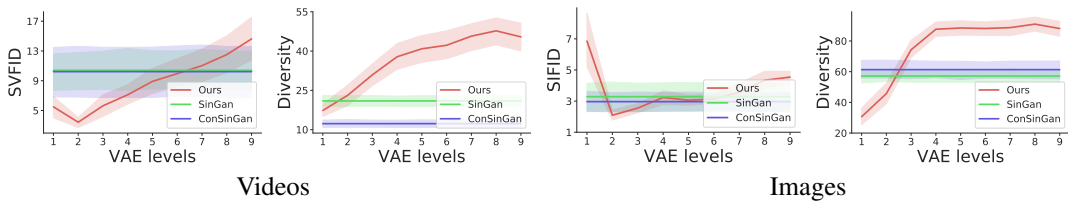

Videos          Images

Figure 5: Diversity vs. Realism. Each graph shows SVFID/SIFID score or Diversity score for different number of VAE levels, as well as for SinGAN [24] and ConSinGAN [28] - represented via straight lines. For SVFID/SIFID - **Lower** is better. For Divesity - **Higher** is better.

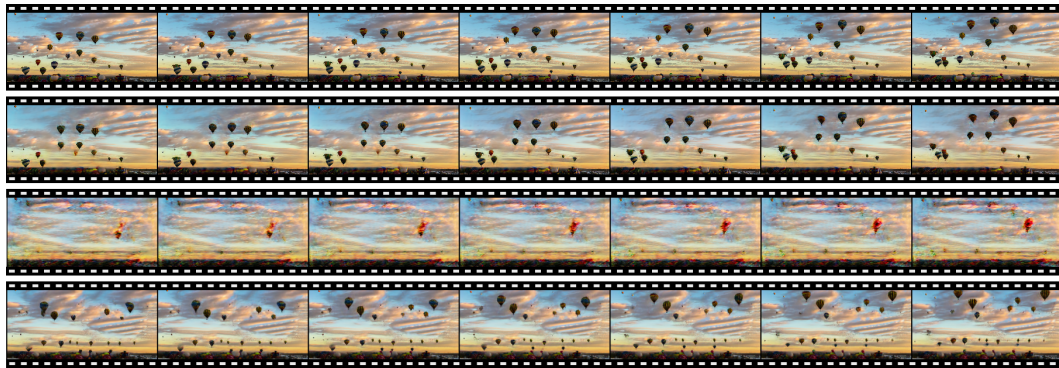

Figure 6: Sample results. **Row 1** - Real sample. **Row 2** - Single VAE level, resulting in high memorization. **Row 3** - Only VAE, resulting in low quality. **Row 4** - three VAE levels and seven GAN levels, resulting in both quality and diversity.

## 4.2  Single-Sample Generation

**Video Generation**    As far as we can ascertain, no other video generation method trains on a single video. We, therefore, compare to the most natural extension of SinGAN [24] and ConSinGAN [28] to videos, by replacing all the 2D convolutions with 3D ones. To evaluate our method, we evaluate the realism (C1) and diversity (C2) of the generated samples. To evaluate (C1), we report the average SVFID score over 50 generated samples for 25 different real training videos. To evaluate (C2) we consider the diversity measure introduced in SinGAN [24]: For a given real training video, we calculate the standard deviation (std) of intensity values (in LAB space) of each pixel over 50 randomly generated videos, average it over all pixels and normalize by the std of intensity values of the training video. This values is averaged over the 25 training videos.

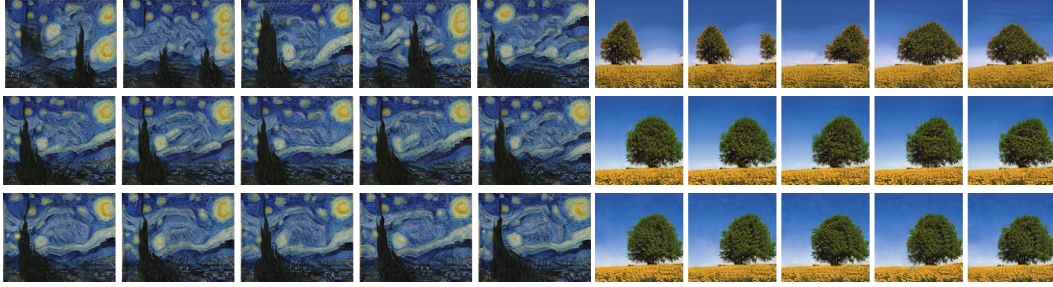

Figure 7: **Image generation** qualitative results, showing five sampled results for two different images. **Top:** Ours, **Middle:** SinGan [24], **Bottom:** ConSinGAN [28].

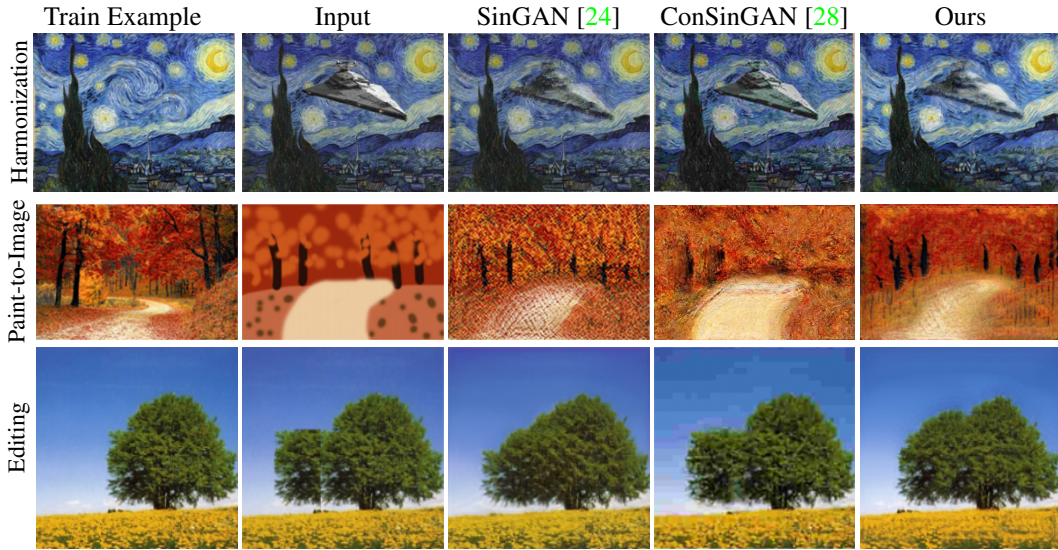

Figure 8: Additional image applications comparison: harmonization, paint-to-image and editing.

Fig. 5 depicts a comparison of our method for different values of $M$ (number of VAE levels) as well as to SinGAN and ConSinGAN ($N$ is fixed to 9). Increasing the number of VAE levels results in more diverse, but less realistic, video samples (higher SVFID value). Using a value of $M$ between 2 and 5, our method is preferable to the baselines both in terms of realism and diversity. A qualitative comparison is shown in Fig. 6. Using a single VAE level results in memorization of the input frames, while using only VAE levels ($M = N = 9$) results in a low quality output. Setting $M = 3$ results in both realistic and diverse frames. The full videos are provided in the SM.

To further evaluate (C1) and (C2), we conducted a user study, comprised of 50 users and 25 real training videos. For each training video, we generate five randomly generated videos from a model trained on this video. The training video is shown alongside the generated videos in a loop. The user is asked to rank from 1 to 5: (C1) How real does the generated video look? (C2) How different do the generated videos look? A mean opinion score is shown in Tab. 3, indicating our method significantly outperforms baselines, both in terms of realness and diversity. Sample videos are provided in the SM. In the SM we also consider the application of training on a (either a short or long) video of a standard $256 \times 192$px resolution and producing random videos of higher $1024 \times 256$px resolution.

**Image Generation**   Our method can also be used for single sample image generation, by replacing 3D convolutions with 2D ones and performing spatial down-sampling and up-sampling. SIFID and diversity scores were calculated as in SinGAN. (C1) and (C2) user studies were also performed, where instead of real videos, 25 real images were used. As can be seen in Tab. 3, our results are more preferable to the baselines, both in terms of realism and diversity. Considering the diversity measure and SIFID score reported in Fig. 5, setting $M = 3$ gives preferable diversity and realism than baselines, but the margin is lower than that of videos. We evaluate our method with four image

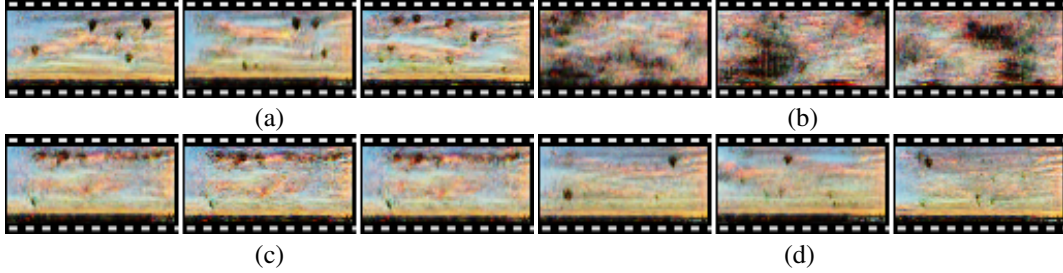

<div align="center">(a)             (b)</div>

<div align="center">(c)             (d)</div>

Figure 9: Alternatives to our patch-VAE for $M = 5$ (showing the output of the patch-VAE part only). Three frames of (a) our method, (b) decoder with $r = 1$, (c) encoder with $r = 1$, (d) Gupta *et al.* [21].

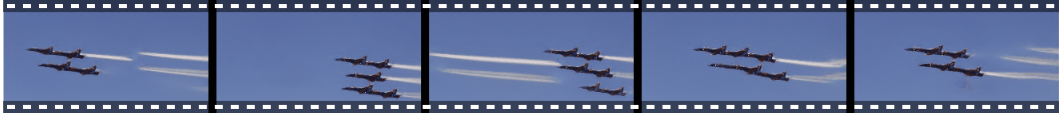

Figure 10: Failure frames of "scene" understanding. A plane's trail appears separate from the plane.

applications, considering generation, paint-to-image, harmonization and editing, as defined by [24]. A qualitative comparison for image generation is provided in Fig. 7, showing higher diversity and quality for our method compared to the baselines. As can be seen in Fig.8 our method performs more realistic harmonization and editing than the baselines, and present a better balance between structure and texture in the paint-to-image task for the given examples. Additional images and comparison to baselines are given in the supplementary material.

**Network freezing** As described in Sec. 3, at scale $n$, we freeze networks $G_1, \ldots, G_{n-1}$. When training the patch-GAN we also freeze $E$ and $G_0$. When training all levels, diversity drops by $51\%$, and SVFID increases by $680\%$. Visually we observe a lot of memorization, as shown in the SM.

**Patch-VAE** As mentioned in Sec. 3, for the patch-VAE formulation, it is important to choose $r$ (the effective receptive field) correctly. In this experiment, we consider only the patch-VAE output after $M := 5$ levels. Our default formulation (i) sets $r$ to 11 (see SM for exact details). We consider the following: (ii) $r$ is set to 1 for all patch-VAE generators $G_0 \ldots, G_M$, by setting all kernels to $1 \times 1 \times 1$, (iii) $r$ is similarly set to 1 for the encoder $E$, (iv) a different formulation of PatchVAE by Gupta *et al.*, [21] (trained on a single sample $x$) is used for $E$ and $G_0$ and the $\mathbb{KL}$ loss term is replaced accordingly (see SM for details). Fig. 9 considers three different frames of a randomly generated sample for each formulation. As can be seen, (ii) results in unrealistic samples, (iii) results in mode collapse and (iv) only generates very few objects at relatively small scale.

**Limitations** Our method is trained in an unsupervised manner on a single input video. As a result, it has no semantic understanding or notion of "scenes". While all the local elements are preserved (people walking, car moving), the global structure may be unnatural: For example, an airplane trail may appear separate from the plane, as can be seen in Fig.10.

## 5 Conclusion

The ability to generate a diverse set of outputs, based on a single structured multi-dimensional sample, relies on the variability that exists within the sample itself. This variability has two aspects. First, there is a structural variability in the relative location of the different parts of the sample. Second, there is a local diversity in the appearance of each part. Image and video patches have been known to repeat in various locations and to exhibit a fractal, multi-scale, behavior. In this work, we model the hierarchical coarse to fine structure of the patches using two complementary technologies. First, to encourage structural diversity, we present a new technique called patch-VAE. Second, to create diversity in details, we employ patch-GANs. These two are combined in a novel hierarchy that is able to surpass the performance of the previous image-based techniques, as well as provide a novel capability of generating diverse videos from a single sample. The same set of tools can be applied more broadly, and we look to apply these for other structural data types, such as hierarchical graphs.

## Acknowledgments

This project has received funding from the European Research Council (ERC) under the European Unions Horizon 2020 research and innovation programme (grant ERC CoG 725974). The contribution of the first authors is part of a Ph.D. thesis research conducted at Tel Aviv University.

## Broader Impact

The ability to create synthetic videos can be used for creating fake videos. Humans tend to perceive media as genuine, which creates a risk. On the other hand, the development of methods such as ours enable the study of methods to detect fake videos.

A positive societal outcome may arise by the privacy-preserving aspect of providing videos in which identifiable elements such as the arrangement of the scene and the people within it have been altered. This can promote privacy, which is an aspect of great societal value as online media sharing continually increases.

## Footnotes

[2]The generative fields are the image synthesis analog of receptive fields, i.e., the spatio-temporal extent that every single column out of the $i$ columns affects.

[3] https://webee.technion.ac.il/people/tomermic/SinGAN/SinGAN.htm

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
