[Supplementary Material]

# Supplementary Material for
# Hierarchical Patch VAE-GAN: Generating Diverse Videos from a Single Sample

## 1  Qualitative Results and Code

Please refer to the attached HTML file (open index.html in a browser) for qualitative results. Our code is also attached in the folder 'code'.

## 2  Implementation Details

### 2.1  Architecture and Training Details

Each generator $G_i$ consists of five $3D$ convolutional blocks. The first $4$ block each consist of a 3D convolution with kernel $3 \times 3 \times 3$ and with padding 1, batch norm and LeakyReLU as layer activation. For the last block, the LeakyReLU is replaced with Tanh activation. This results in an effective receptive field $r$ of 11.

$E$ consists of a similar architecture to $G_i$, except we replace Batch-Normalization with Spectral-Normalizatiom [1] and all blocks use LeakyReLU activation. $M$ and $S$ are each a separate convolutional layer with kernel $3 \times 3 \times 3$ and padding 1 (no activation is used).

The patch discriminator $D_i$ follows the same architecture as $G_i$, where we replace Batch-Normalization with Spectral-Normalization, and the final layer has no activation. We note that all the convolutional blocks preserve the input dimension (height, width and time).

The discriminators $D_i$, generators $G_i$ and encoder $E$ each use 64 channels for each block. $M$ and $S$ each consist of a single convolutional layer which increases the 64 input channels to 128, and so the dimension of $z_i$ as defined in Sec. 3.1, is 128. As in the convolutional block, $M$ and $S$ use a $3 \times 3 \times 3$ kernel and padding 1.

As noted in Sec. 4, unless otherwise stated, we set $N = 9$ and $M = 3$. Input videos are of 24 FPS. Down-sampling and up-sampling is as described in Sec. 3.2, with the 4 frames used at scale 0 and 13 frames at scale $N$. At scale 0, we set the height to 32px while at scale $N$ we set it to 256px. The height at scale $n$ is set to be 1.33 the height of scale $n-1$, or to 256px, whichever is lower.

We use an Adam optimizer with learning rate of $5 \times 10^{-4}$ for each scale. Moving to the next scale, we decay the learning rate of $E$ and $G_0$ at a rate of 0.2 per scale, for the duration of the VAE training. The $\mathbb{KL}$ weights $\beta_{vae}$ and $\beta_{adv}$ are set to 0.1. We train on a single Nvidia 2080 Ti, for 20k iterations per scale. Before training at scale $n > 0$, we initialize $G_n$ (resp. $D_n$) with the weights of $G_{n-1}$ (resp. $D_{n-1}$).

### 2.2  SVFID Measure

SinGAN [2] showed that SIFID captures how fake generated images look compared to the real training sample. In our case, instead of a pre-trained Inception network [3], we consider C3D [4] network pretrained for action recognition on Sports-1M dataset [5] and fine-tuned on UCF-101 [6]. Given our real and fake video samples, we first pass them through the fully convolutional part of C3D

resulting in a spatio-temporal activation map of 512 channels. We then take the FID [7] between the 512-sized vectors (one for each location in the map) of our real and fake sample. We note that baseline methods produced 16 frames instead of our 13 frames. Spatially, MoCoGAN [8] and TGAN [9] produce $64 \times 64$ output and TGAN-v2 [10] produces a $128 \times 128$ output, while our output is of height 256px (keeping the aspect ratio of original video). We therefore preformed a trilinear interpolation for all videos to a spatial dimension of $112 \times 112$ and temporal dimension of 16 before calculating the SVFID score.

### 2.3 FID

As mention in Sec. 4.1, we calculate the Frechet Inception Distance [7] (FID score) between 50 generated samples, and the test set of UCF-101. Initial video resizing is done as for SVFID score. We note that the embedding size used to calculate the FID metric is 512. As this is much larger then the number of generated samples used for the FID calculation, the rank of the estimated co-variance matrix in the FID calculation is smaller than its dimension. This results in an inaccurate FID estimation. Therefore, for 50 independent trials, we select 50 features from the 512 feature vector at random, compute the FID score using these features, and average the FID score over all trials. As this computation is done differently to other works, FID score should be considered in comparison to baselines and not in comparison to FID score reported in other works.

### 2.4 PatchVAE by Gupta *et al.*, [11] baseline

In Sec. 4.2, the PatchVAE baseline by Gupta *et al.*, [11] is considered. We follow the same procedure described in Gupta *et al.* [11], decomposing the $\mathbb{KL}$ loss to a $\mathbb{KL}$ loss between a Bernoulli prior and a Gaussian prior. We use a latent dimension of 128 as is done in our method. Refer to Gupta *et al.* [11] for additional details.