[Reviews · NeurIPS 2020]

Review 1

Summary and Contributions: This paper presents a method to compute unconditional video generation, conditioned on a single input video. Technically, it adds a patch-VAEs to the low levels of SinGAN. This allows for greater diversity, while the latter GAN levels allow for more realistic sample generation.

Strengths: + The results are compelling and seem better than the prior methods. + The evaluation is well done, it includes a user study, as well as qualitative results, and quantitative comparison to baselines over several datasets. + Good intuition as to what the impact of N is. + The paper is well written and easy to follow + Results are shown for the more general single image applications and seem to indicate similarly improved diversity. - The paper presents a relatively small modification to SinGAN. - Some evaluation of this idea for the single image application is made, but it is not conclusive. - r does not seem to be studied well. I can see that r=1 fails, but it is not clear to me what happens when r goes much bigger than 11 for example. - low resolution, low fps and short videos are somewhat hard to evaluate (you have to squint a bit).

Weaknesses: - The paper presents a relatively small modification to SinGAN. - Some evaluation of this idea for the single image application is made, but it is not conclusive. - r does not seem to be studied well. I can see that r=1 fails, but it is not clear to me what happens when r goes much bigger than 11 for example. - low resolution, low fps and short videos are somewhat hard to evaluate (you have to squint a bit).

Correctness: As far as I can tell.

Clarity: The paper is well written and easy to follow.

Relation to Prior Work: As far as I can tell.

Reproducibility: Yes

Additional Feedback: In summary, this work provides a study into improving the diversity of SinGAN and CoSinGAN using a VAE at low levels of the image/video generation pyramid. The novelty of this idea is modest, but the paper is well written and I believe would give insight into this problem domain to a typical reader. In addition to the concerns mentioned in the weakness section, I have a few minor comments/questions. In Fig6, row 3 (the example with only a VAE for all levels), why is there no diversity? This does not match the plots in Fig5, or my understanding from reading the paper. It would be very nice to see higher framerates (even at the same overall length) in these generated videos, what is the main limitation that prevents this? Maybe a frame interpolation method could be used, just for visualization purposes.


Review 2

Summary and Contributions: The paper suggests learning a generative model from a single short video example. This is done by gradually training a multi-scale pyramid of networks, which contains patch-VAEs at coarser scales (to increase variability) and patch-GANs at the finer scales (to increase visual quality). At test time, this pyramid is able to generate new diverse video samples. The authors compare the quality and diversity of the generated videos with both externally trained and internal methods and show the suggested approach outperforms baselines in both aspects.

Strengths: This is the first attempt to learn a generative model from a single short video, which is a challenging task due to the strong temporal relations among the frames. Evaluation seems comprehensive and includes quantitative, qualitative, and user-study comparisons to both external video generation approaches and internal single-image baselines which were extended to be suitable for videos. The paper also includes an analysis of the network design (number of VAE scales). In addition, the authors also demonstrate how their method can be easily trained for images instead of videos. The idea of combining VAE and GAN in a multi-scale fashion is interesting and seems to enable control over the tradeoff between variability and visual quality.

Weaknesses: The motivation of the work is not fully clear. In the way the problem is presented now, it seems like the main motivation is to extend single image GANs (e.g. SinGAN) to videos. However, SinGAN is able to solve a wide variety of applications, which the presented method doesn’t attempt to solve. The generated videos are many times not realistic (e.g. air-balloons are gradually transferred into sky along frames). In addition, the method is illustrated on simple videos with non complex motion. I wonder how the method works for more complicated motion which includes both a moving camera, and objects moving at different directions. Furthermore, the method is demonstrated for only extremely short videos (13 frames in a 24 FPS video is approximately 0.5 sec).

Correctness: Yes

Clarity: L169-187 needs to be clarified. Current explanations are cumbersome and are difficult to follow. In particular the difference in the generation of the reconstruction sample and random samples should be better explained. L122-126: the intention here is not fully clear. SinGAN can also be trained with a few examples with mean over the loss. Where is the difference?

Relation to Prior Work: In Sec. 4.1, the comparison with externally trained methods is a bit strange. (i) Visual examples of all the external methods seems very poor and perhaps cherry-picked. (ii) The variability of the externally trained methods is obviously much larger, this comparison should be included or at list mentioned.

Reproducibility: Yes

Additional Feedback: Eq. 10: this is not a WGAN-GP loss: there is no 0,1 maximization/minimization, and the gradient penalty is calculated in addition to the adversarial loss and not on top of it. Is this a typo? If not, what is the motivation for using this loss? Some details regarding the user-study are missing: Were videos played for unlimited/limited time? In contrast to regular encoder-decoder architecture, here the encoder doesn’t reduce the dimensionality in order to get latent representation (z and x_0 are of the same dimension). Is there a reason for this design choice? It is a bit strange to me that when training the N-th scales G_0 keep changes, but all the other generators (0<n<N) are fixed. The fixed generators were trained with G_0 and not it is now being updated so their training does not necessarily hold anymore. Potential typos: L175: \hat{x}^n should be \bar{x}^n L177: I believe \bar{x}^M should also be random (i.e. generated with random z and not with E(x_0)) ------------------------------------------------------- Post-rebuttal: I have read the rebuttal and the other reviews. I am still concerned about the (lack of) motivation for learning a generative model from a single short video. The authors don't provide any application or use-case for such a model (in the rebuttal they only mention applications for images, not videos). Generative models that are trained on a dataset of images/videos are known to be useful for solving many tasks. But when training on a single example, it is not a-priori clear what such a model can be good for. Showing that the model can generate random samples is nice, but that's not an application. I can think of potential use-cases, like video editing/inpainting/temporal-super-resolution, etc. But it's not clear that the proposed model can easily support any of these applications, and in any case none of these are mentioned in the paper or rebuttal. This, together with the fact that the videos are very short and low-res, make me think that this work is a bit premature for publication. I therefore keep my initial score.


Review 3

Summary and Contributions: This work proposes a method based on a hierarchy of patch-based VAEs and GANs for generating videos from a single video. The first part of the hierarchy consists of patch-VAEs to introduce diversity and avoid mode collapse. The second part consists of patch-GANs to add finer details and better quality to the generated videos. The study has been inspired by a previous work SinGAN, which uses patch-GAN. Based on the authors' claim, this study is the first work capable of unconditional video generation from a single video. The contributions of the authors are introducing and formulating a patch-VAE for a single sample generation, and combining patch-VAE and patch-GAN in a hierarchy to produce diverse and high-quality videos from a single training sample.

Strengths: The idea of the hierarchy of VAEs and GANs for ensuring both diversity and quality is very interesting, if it has not been introduced before. For the evaluation, the paper compares its proposed method with general video generatio methods, and the extension of image generation methods from a single image to the video setup. The authors show that their method outperforms the other methods. This work could be very useful for data augmentation, since it can augment each video sample with diverse and novel sam

Weaknesses: Based on figure 2, the input to the encoder during the training process is the low-resolution version of the input (x0). Since it is not a super resolution setup, It is not clear why authors have not used the full-resolution input, since it might contribute to the latent space. The notations used in equation 7 and 8 could be better. For example, in equation 7, the left-hand side implies that loss is between X0 and Xn, however, it is a reconstruction loss between Xn and X_bar_n. The code of the video generation work [31] can be found at https://github.com/musikisomorphie/swd. Its related conference version [a] is published at CVPR 2019. It would be better if the authors can cite it as well. Since this paper also exploits incrementally increases both the spatial and temporal resolution during training, I suggest to compare this method if possible. [a] Wu et al., Sliced Wasserstein Generative Models. In CVPR 2019. To compare the proposed methods with other video generation methods, the authors explain in section 4.1 that they generated samples from the baselines, find the two nearest neighbors in the dataset, and train their method on the first one. Then the paper computes the evaluation metric between the generated videos and the 2 nearest neighbors for both their method and the baseline. Although the authors argue that only evaluation metric between the generated video and the first NN in the dataset i in favor of their method (because the method is trained on that sample), considering UCF-101 dataset, lots of video samples are short pieces of a longer video, so it is very likely that the second NN also comes from the same source video that the method is trained on, so it is expected that their method still be more similar to the second NN comparing to the generated sample from video generation methods, which are not overfitted on angle video.

Correctness: While I did not go through all the details, the claims, method and empirical methodology seem correct.

Clarity: In general, the paper is written well.

Relation to Prior Work: Yes, the discussion looks clear to me.

Reproducibility: Yes

Additional Feedback: ############post rebuttal comment################## I share the same concern with the other reviewers that this work lacks a good study on potential applications. Nevertheless, I really like the idea of hierarchical patch VAE-GAN and its promising visual results, and it seems a valuable attempt to promote the research on unsupervised video generation. The rebuttal also addresses my other concerns. Therefore, I tend to maintain my score.


Review 4

Summary and Contributions: The paper proposes a patch level hierarchical method involving multi-scales of VAE followed by multi-scales of GAN to generate a diverse set of video samples from a single video clip. The work is in the spirit of SinGAN that deals with generating new images from a single image. The method produces very impressive videos both in terms of realism as well as diversity when compared to strong baselines.

Strengths: 1. The paper is well-written and the approach proposed and it's several variants are clearly explained. The main approach involves two components: a patch-VAE and a patch-GAN. The patch-VAE consists of a single encoder and a series of several decoders at multiple levels, M with increasing granularity. This output of patch-VAE is fed to patch-GAN that consists of a series of 3D GANs at multiple levels, N with increasing granularity. The best variant has M = 3 and N = 9. They compare with their own two variants, a) pure patch-VAE (i.e only VAE at all levels) and b) a single VAE level (i.e traditional VAE) and show that a) results in low quality samples and b) results in high memorization. 2. To the best of my knowledge, this is the first work that attempts to generate diverse and novel videos from a single video. The method is also compared with the 3D variants of SinGAN and ConSinGAN that were originally proposed for images. 3. The experiments conducted on UCF-101 and a small subset of Youtube 8M dataset are quite thorough. The proposed approach beats all baselines, both in terms of SVFID (a metric devised in the paper to evaluate the quality and diversity of novel videos from a single-video) as well as user-preference study.

Weaknesses: 1. There isn't any discussion on what kind of videos the method would fail on. How would the method generalize, say cityscapes video or Mapillary dataset which are traffic scene datasets? It will be interesting to know the limitations of the method so it gives a direction for the research community in terms of the open problems. 2. The authors should also compare with the more latest multiple sample video generation methods like Villegas et al [1], for example. Although I don't expect these methods to beat the proposed approach, it is important to compare with a few and cite a few more latest ones. References: 1] High Fidelity Video Prediction with Large Stochastic Recurrent Neural Networks, Villegas et al, Neurips 2019

Correctness: Yes

Clarity: Yes

Relation to Prior Work: Yes

Reproducibility: Yes

Additional Feedback: I'm happy with author's responses to the questions raised by the reviewers. I think all of us acknowledge that the videos on which the method is evaluated involve simple scenes and a uni-directional motion. But given that this line of research (training on single video) is at a nascent stage, I support it's acceptance. I encourage the authors to include a limitation section highlighting the kind of videos the methods fails on and why that is the case. It will only help other researchers in the community for new directions in this line of research and perhaps increase the work's impact.

[Author Response · NeurIPS 2020]

Thank you for the insightful comments. We will make sure to address all requests for elucidation in the revision.

**R1 and R2: Single image evaluation:** Our main motivation is that of single sample video generation. However,
as discussed in Sec. 4.2, our method also applies to images. We follow a similar evaluation protocol to SinGAN:
quality (SIFID), diversity (Fig. 5 RHS) and a user study (Tab. 3 RHS). Using $M = 3$ (see L.245-252), we outperform
with all these metrics. **Additional applications:** In Fig. (a-c) below, we consider the applications of inpainting,
harmonization and editing, as used in SinGAN. We also consider the following application: produce random videos of
higher $1024 \times 256$px resolution, compared to the $256 \times 192$px resolution of training video. Two consecutive frames are
shown for two videos ($d$ and $e$). Full videos will be provided. **Low resolution, FPS and length:** We produce videos of
the same FPS as the training video (24), and of a standard 256px resolution. For visualization purposes, GIFs produced
in the SM are played at a slightly lower FPS. By using an input of higher FPS, videos of higher FPS can be produced.
Our method produces 13 frames as opposed to baselines' 16, but of higher resolution. This is a result of GPU memory
limit and sampling technique, and not a limitation of the method. See Sec. 3.2 and SM Sec. 2.1 for additional details.

**R1: Many differences to SinGAN:** We introduce a novel formulation of a VAE that operates on patches of a single
sample. Previously, only patch-GAN existed [25, 24, 26, 27]. While attempts were made to combine GANs and VAEs
[17, 18, 19,20], we are the first to do so within an hierarchical structure. We believe that both of these formulations
could be used in other generation tasks. Moreover, we consider the challenging task of diverse video generation from
a single sample. SinGAN completely fails on this task, and collapses, even when modified accordingly. **Study of $r$:**
Following the review, we conducted an additional experiment with $r = 31$. As noted in L.120-121, setting $r$ too large,
results in a small number of samples $K$. In our experiment this resulted in failure to generate realistic samples. **Fig 6,**
**row 3:** Fig 5 shows that when using the unoptimal setting of $M = 9$ (only VAE), SVFID score is high, indicating low
quality, yet diversity is high. In Fig. 6, row 3, the sky is scattered completely differently from row 1. The appearance of
only a few balloons is specific to this randomly generated sample. In the SM, for example, many balloons appear.

**R2: Realism:** While video quality can be improved, our method significantly outperforms baselines (see Fig. 5 and
Tabs. 1-3). **Complex motion:** Note first SM example in 'Longer Training Videos' depicting hot-air balloons moving in
different directions and first SM example in 'Randomly Generated Videos' of skydivers depicting significant camera
motion. Following the review, we considered an input video of a savanna, with cheetahs and an impala moving at
different directions, and with a moving camera. Fig. (f) shows a randomly generated output. Random videos capture
the same camera motion of the input video, while generating realistic objects (cheetahs and impalas) moving in novel
directions. **L.122-126:** Applying SinGAN on multiple samples would require reconstructing each sample, at the the
coarsest level, from the same fixed noize $z^*$, resulting in bad reconstructions. As our method uses a VAE formulation,
it does not have this limitation. **Prior Work:** Random samples were generated using the public implementation of
baseline methods. We note that MoCoGAN [31] did not consider the UCF101 dataset and hence resulted in the worst
generation quality. Regarding variability, we do not claim to generate variable content from multiple videos, as may be
the case for multiple-sample baselines. However, our method is superior to baselines in generating diverse content from
the same internal statistics of the input video (see Tab. 3 and Fig. 5). We will clarify this in the next revision. **Eq. 10:**
We thank the reviewer for noting this typo, which we will correct. A WGAN-GP loss with gradient penalty is used.
**User study videos** are shown for unlimited time. **Latent representation dimensionality**: As discussed in L.100-104,
the encoder $E$ can be seen as creating a distribution $q(z|x)$ of of r-sized patches. Our default setting uses $r = 11$, and
$c = 128$, and so the dimensionality of a patch of size 3x11x11x11 in the input video is reduced to 128. **Change of**
$G_0$: Unlike $G_1, \ldots, G_n$, $G_0$ has a special role of decoding a video from noize. $G_0$ adapts to the distribution $q(z|x)$ of
the latent encodings (which may change when moving to a new level of the hierarchy) produced by the encoder. The
distribution of lower resolution inputs to $G_1, \ldots, G_n$ is fixed, and so $G_1, \ldots, G_n$ do not continue training (L.157-159).

**R3 and R4: Comparison:** We will gladly add a discussion of [1] and of [a], both of which operate in the multiple
sample setting. We did our best to compare with methods for which code was made public. For [1, 32] this is not the
case. Regarding [31], we thank R3 for pointing to the implementation. We will make an effort to add this comparison.

**R3: Encoder input:** As our encoder $E$ and decoder $G_0$ are fully convolutional, $G_0$'s output is of the same size as $E$'s
input, which must match $x_0$'s size. Instead, to enrich the latent space using $x_n$'s finer details, $E$ is still updated through
$x_n$'s reconstruction loss. **Multiple sample baseline:** We verified that for the subset of videos where the 2nd NN is
from a completely different video, SVFID is indeed somewhat higher, but still superior by a sizable gap to baselines.

**R4: Limitations:** Our method is trained in an unsupervised manner on a single input video. As a result, it has no
semantic understanding or notion of "scenes". While all the local elements are preserved (people walking, car moving),
the global structure may be unnatural: For example, in the SM, example 13 of airplanes, the plane trail appears separated
from the plane in the first random example. We will add a discussion of the limitations in the next revision, specifically
discussing the inability to extract complex spatial relations as those in Cityscapes from only one video.



[Meta-Review · NeurIPS 2020]

The paper studies GAN training from a short video clip. Reviewers appreciated the challenges of the task, the fact that it hasn't been attempted in this form and also the combination of VAE and GAN. Reviewers were concerned about the use cases, realism, simplicity of the motion and length of the videos. These concerns remained after reading rebuttal. Three reviewers valued novelty of the task more strongly and AC concurs. This will be a baseline for future improvements.